# The leukocyte non-coding RNA landscape in critically ill patients with sepsis

Brendon P Scicluna[1,2]*, Fabrice Uhel[1], Lonneke A van Vught[1], Maryse A Wiewel[1], Arie J Hoogendijk[1], Ingelore Baessman[3], Marek Franitza[3], Peter Nürnberg[3,4], Janneke Horn[5], Olaf L Cremer[6], Marc J Bonten[7,8], Marcus J Schultz[5], Tom van der Poll[1,9], Molecular Diagnosis and Risk Stratification in Sepsis (MARS) consortium

[1]Amsterdam UMC, University of Amsterdam, Center for Experimental Molecular Medicine, Amsterdam Infection & Immunity, Amsterdam, Netherlands; [2]Amsterdam UMC, University of Amsterdam, Department of Clinical Epidemiology, Biostatistics and Bioinformatics, Amsterdam, Netherlands; [3]Cologne Center for Genomics, University of Cologne, Cologne, Germany; [4]Center for Molecular Medicine Cologne, University of Cologne, Cologne, Germany; [5]Amsterdam UMC, University of Amsterdam, Department of Intensive Care Medicine, Amsterdam, Netherlands; [6]Department of Intensive Care, University Medical Center Utrecht, Utrecht, Netherlands; [7]Department of Medical Microbiology, University Medical Center Utrecht, Utrecht, Netherlands; [8]Julius Center for Health Sciences and Primary Care, University Medical Center Utrecht, Utrecht, Netherlands; [9]Amsterdam UMC, University of Amsterdam, Division of Infectious Diseases, Amsterdam, Netherlands

**Abstract** The extent of non-coding RNA alterations in patients with sepsis and their relationship to clinical characteristics, soluble mediators of the host response to infection, as well as an advocated in vivo model of acute systemic inflammation is unknown. Here we obtained whole blood from 156 patients with sepsis and 82 healthy subjects among whom eight were challenged with lipopolysaccharide in a clinically controlled setting (human endotoxemia). Via next-generation microarray analysis of leukocyte RNA we found that long non-coding RNA and, to a lesser extent, small non-coding RNA were significantly altered in sepsis relative to health. Long non-coding RNA expression, but not small non-coding RNA, was largely recapitulated in human endotoxemia. Integrating RNA profiles and plasma protein levels revealed known as well as previously unobserved pathways, including non-sensory olfactory receptor activity. We provide a benchmark dissection of the blood leukocyte 'regulome' that can facilitate prioritization of future functional studies.

*For correspondence:
b.scicluna@amc.uva.nl

Competing interests: The authors declare that no competing interests exist.

## Introduction

Sepsis is a multifaceted syndrome that develops as the consequence of an abnormal host response to infection leading to organ failure and high risk of death (*Angus and van der Poll, 2013*; *Cecconi et al., 2018*). It is estimated that 2–5 million deaths worldwide are attributable to sepsis (*Fleischmann et al., 2016*). Despite empirical antimicrobial therapy and advances in intensive care, it is expected that sepsis will remain a major healthcare problem. As such, sepsis has been recognized as a global health priority in 2017 by the World Health Assembly and WHO (*World Health Organization, 2017*). In spite of more than 100 clinical trials having evaluated drugs targeting specific components of the host response to infection (*Marshall, 2014*), no specific treatment for sepsis has been approved (*Angus and van der Poll, 2013*; *Cecconi et al., 2018*). This argues for a deeper

understanding of sepsis immunopathology to identify veritable drug targets (*Marshall, 2014*; *Tse, 2013*).

Protein-coding RNA expression profiling of blood leukocytes from sepsis patients has helped to broaden our understanding of sepsis immunopathology (*van der Poll et al., 2017*), for example, by unmasking defects in leukocyte energy metabolism of sepsis patients (*Cheng et al., 2016*), and by classifying sepsis patients as transcriptomic endotypes with prognostic and pathophysiological value (*Scicluna et al., 2017*; *Davenport et al., 2016*; *Wong et al., 2009*). From fruit flies to man, the protein-coding part of genomes from different species is remarkably similar in numbers and functions (*Liu et al., 2013*), which suggests that numerous aspects of complex biology in eukaryotes might stem from non-protein-coding regions of the genome. The increase in genomic coverage of tiled microarrays and massive cDNA sequencing undertaken by the Functional Annotation of the Mammalian genome (FANTOM) consortium revealed pervasive transcription outside of the known gene loci (*Kapranov et al., 2002*; *Carninci et al., 2005*). Moreover, such studies facilitated the demonstration that non-coding RNAs were under negative evolutionary selection, which implied functionality rather than plain 'transcriptional noise' (*Ponjavic et al., 2007*). Indeed, a substantial proportion of non-coding RNA, by general convention defined as long (>200 nucleotides) or small (<200 nucleotides) non-coding RNAs, yields clear phenotypic effects in both in vitro and in vivo functional studies (*Zhu et al., 2016*; *Gebert and MacRae, 2019*; *Atianand et al., 2016*; *Carpenter et al., 2013*). Ever-growing numbers of small non-coding RNAs, for example micro (mi) RNAs (20–24 nucleotides), or long non-coding RNAs such as long intergenic non-coding (linc)RNAs, have been linked to human diseases (*Bao et al., 2019*; *Esteller, 2011*). An important aspect of non-coding RNAs is their capacity for precise regulation of cellular biological processes via epigenetic mechanisms, including complex immune system processes (*Carpenter, 2018*; *Atianand and Fitzgerald, 2014*; *Mehta and Baltimore, 2016*).

Knowledge of the non-coding RNA landscape in patients with sepsis is limited. Here we report a comprehensive screen of non-coding RNA expression patterns in blood leukocytes of patients with sepsis and their relation to clinical characteristics and soluble mediators of the host response. In addition, by using a guilt-by-association approach we positioned non-coding RNAs in network modules encompassing protein-coding RNA reflecting distinct cellular biological pathways.

## Results

### Protein-coding and non-coding blood transcriptomes

In order to build a comprehensive map of RNA expression in the context of sepsis, we evaluated protein-coding, long and small non-coding RNA expression in whole blood leukocytes from 156 sepsis patients and 82 healthy subjects (median age (Q1–Q3), 54 (42 – 60); 26% male). Patient characteristics are tabulated in *Table 1*, causative pathogens in *Supplementary file 1*. Principal component (PC) analysis of the most abundant protein-coding RNAs (n = 18,063) and long non-coding RNAs (n = 16,087) showed clear partitioning of patients with sepsis distinct from healthy subjects (*Figure 1A*). In contrast, small non-coding RNAs (n = 4949) showed only minimal separation between patients and healthy subjects. We observed similar patterns after calculating the molecular distance to health (MDTH)(*Berry et al., 2010*; *Dunning et al., 2018*) index, a measure of transcript-level expression perturbation relative to health, with significantly higher MDTH indices in sepsis (*Figure 1B*). Notably, long non-coding RNA transcripts exhibited the broadest expression perturbations in healthy participants and sepsis patients, exemplified by the highest overall MDTH indices (*Figure 1B*).

Comparing sepsis patients to healthy subjects identified 15,097, 13,158, and 635 significantly altered (adjusted p-value <0.01) protein-coding, long non-coding, and small non-coding RNAs, respectively (*Figure 1C*). Ingenuity pathway analysis of the significantly altered protein-coding RNA transcripts revealed associations with various canonical signaling pathways that included elevated pro- and anti-inflammatory pathways, cell cycle, DNA damage response, and metabolic pathways (*Figure 1—figure supplement 1*). Transcripts with reduced expression were predominantly associated with T helper cell activation, antigen presentation, and B cell responses. Results on protein-coding RNA profiles are in agreement with previous reports from our and other groups (*van der Poll et al., 2017*). LincRNAs, antisense, and pseudogene RNA transcripts represented the most highly

**Table 1.** Baseline characteristics and outcomes of critically ill patients with sepsis.

| Parameter | Sepsis patients (n = 156) |
|---|---|
| Age, years | 62 (50 - 70) |
| Male sex | 98 (62.8) |
| White ethnicity | 140 (89.7) |
| Medical admission | 117 (75.0) |
| Immune suppression | 45 (28.8) |
| Cardiovascular insufficiency | 43 (27.6) |
| Malignancy | 45 (28.8) |
| Renal insufficiency | 18 (11.5) |
| Respiratory insufficiency | 37 (23.7) |
| Charlson comorbidity index | 4 (2 - 6) |
| APACHE IV score | 72 [58 - 92] |
| SOFA score | 7 (4 - 9) |
| Shock | 86 (55.1) |
| Mechanical ventilation | 128 (82.1) |
| *Primary diagnosis* | |
| Pneumonia | 99 (63.5) |
| Community-acquired | 68 (43.6) |
| Hospital-acquired | 31 (19.9) |
| Abdominal sepsis | 57 (36.5) |
| *Outcome* | |
| 28-day mortality | 48 (30.8) |
| 90-day mortality | 59 (37.8) |
| 1-year mortality | 77 (49.4) |

Data presented as median [Q1–Q3], or n (%).

Abbreviations: APACHE, Acute Physiology and Chronic Health Evaluation; ICU, Intensive care unit; GI, gastrointestinal; SOFA, Sequential Organ Failure Assessment.

The online version of this article includes the following source data for Table 1:

Source data 1. Baseline characteristics and outcomes of critically ill patients with sepsis.

altered long non-coding RNA biotypes in sepsis relative to health (*Figure 1D*). Micro (mi)RNAs, stem loop RNAs, and small nucleolar (sno)RNAs were the most abundant small non-coding RNA biotypes (*Figure 1E*).

## Protein-coding and non-coding blood transcriptomes, demographics, and clinical characteristics

In order to understand inter-individual variation in RNA expression profiles, we set out to determine the contribution of demographics and clinical characteristics to protein-coding and non-coding RNA expression variation in sepsis patients (*Figure 2*), as well as healthy subjects. Using a variance partition (multivariable) approach (*Hoffman and Schadt, 2016*), differences in gender and age of healthy subjects explained 5%, 4%, and 4% of the variation in protein-coding, long non-coding, and small non-coding RNA expression, respectively (*Figure 2—figure supplement 1A*). Specific transcripts had high percentages of explainable variance, in particular long non-coding RNAs against gender. Not surprisingly, expression of long non-coding RNAs positioned on the X and Y chromosomes, for example *TXLNGY*, *LINC00278*, and *XIST* had 98%, 97%, and 94% of variance explained by gender, respectively (*Figure 2—figure supplement 1B*). In sepsis patients, a multivariable model that incorporated demographics and common clinical characteristics, including APACHE IV, SOFA scores,

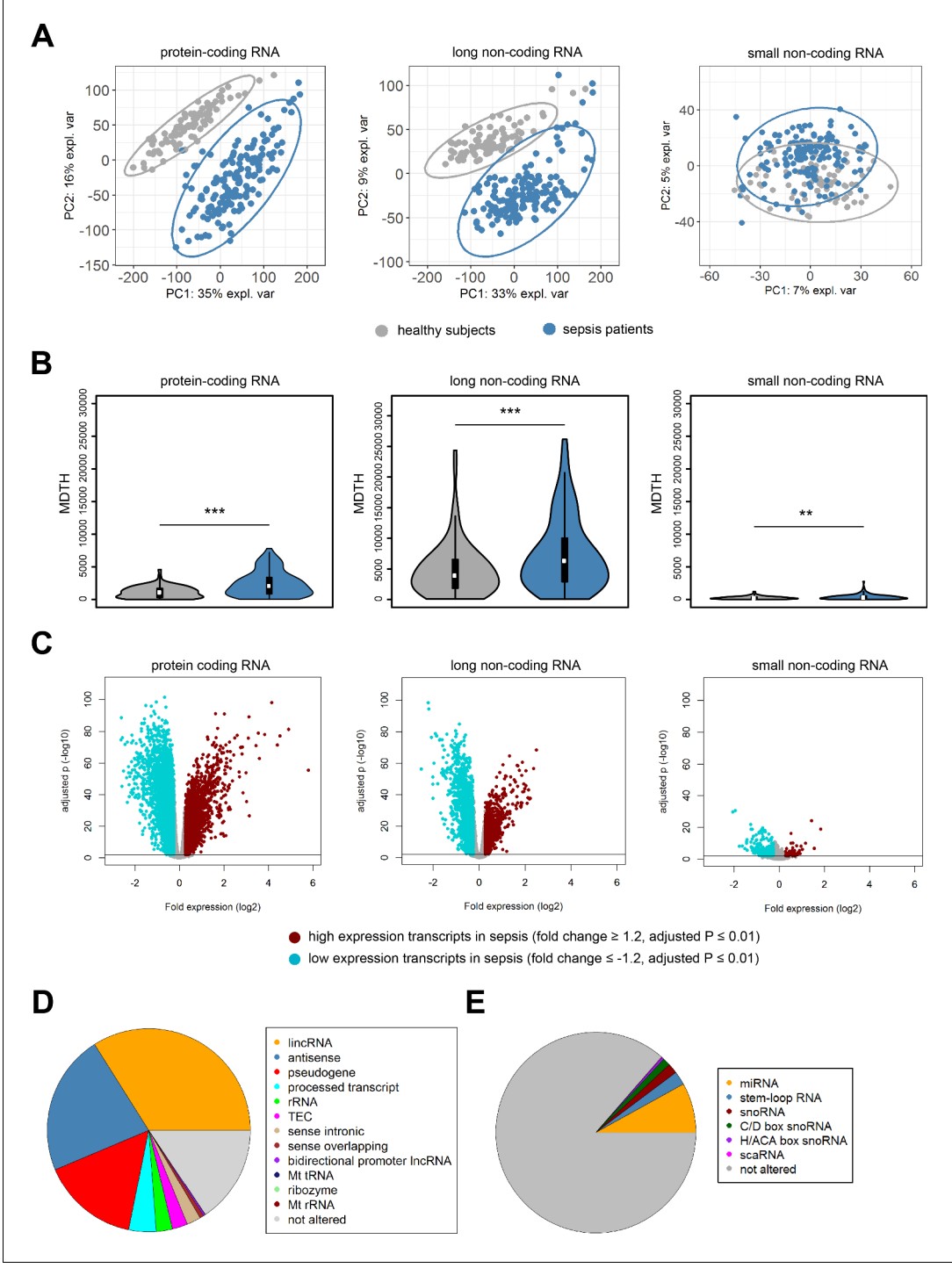

**Figure 1.** Coding and non-coding RNA expression in leukocytes of sepsis patients and healthy individuals. (**A**) Principal component (PC) plot depicting PC1 and PC2, and (**B**) the molecular distance to health (MDTH) index of protein-coding (n = 18,063), long non-coding (n = 16,087), and small non-coding RNAs (n = 4949) in healthy subjects and sepsis patients. \*\*p<0.01; \*\*\*p<0.001. (**C**) Volcano plot representation of differences in coding and non-coding RNA expression between sepsis patients and healthy subjects. Horizontal (black) line denotes −log10 transformed adjusted p-value of 0.01. (**D**) Pie chart showing the subclass distribution of significant long non-coding RNA (adjusted p<0.01). LincRNA, long intergenic non-coding RNA; rRNA, ribosomal RNA; TEC, to be experimentally confirmed; Mt tRNA, mitochondrial transfer RNA; Mt rRNA, mitochondrial ribosomal RNA. (**E**) Pie chart showing the subclass distribution of significant small non-coding RNA (adjusted p<0.01). miRNA, microRNA;

*Figure 1 continued on next page*

*Figure 1 continued*

snoRNA, small nucleolar RNA; C/D box snoRNA, C/D box small nucleolar RNA; H/ACA box snoRNA, H/ACA box small nucleolar RNA; scaRNA, small cajal body-specific RNA.

The online version of this article includes the following figure supplement(s) for figure 1:

**Figure supplement 1.** Ingenuity pathway analysis of significant protein-coding RNA in sepsis relative to health.

shock and Charlson comorbidity indices, cumulatively explained 18%, 13%, and 8% of protein-coding, long non-coding, and small non-coding RNA expression variance, respectively (*Figure 2A*). Specifically, sepsis primary site of infection (lung or abdomen) and place of acquisition (community or hospital) explained the highest proportion of variation in protein-coding (6.7%) and long non-coding (4.4%) RNA expression (*Figure 2A*). Despite overall low proportions of variance explained, outlier RNA transcripts could be detected. For example, some specific transcripts demonstrated high individual explained variance against primary sepsis diagnosis, including protein-coding RNA encoding basic leucine zipper and W2 domains 1 (BZW1); long non-coding RNA SUMO2 pseudogene 1 (SUMO2P1); and small non-coding RNA miRNA hsa-miR-7855–5 p (*Figure 2B*). Septic shock explained low proportions of variation in RNA expression (*Figure 2A*), and directly comparing patients with septic shock to patients without shock resulted in 837 and 80 significantly altered protein-coding and long non-coding RNA, respectively (*Figure 2C*). High expression protein-coding RNA included matrix metalloproteinase 8 (*MMP8*), resistin (*RETN*), and lipocalin 2 (*LCN2*). Low expression protein-coding RNA included a Na+/Ca2+ exchanger (*SLC8A1*), membrane metalloendopeptidase (*MME*), and interleukin (IL-) six receptor (*IL6R*). Long non-coding RNA included lincRNA lung cancer-associated transcript 1 (*LUCAT1*; low expression) and antisense RNA (*LRRC75A-AS1*; high expression) (*Figure 2C*). No significant alterations were identified in small non-coding RNA expression profiles. Evaluating RNA expression in patients discordant for survival after 28 days identified 146 significantly altered protein-coding RNA (*Figure 2—figure supplement 1C*). No significant differences were uncovered in non-coding RNA expression profiles, suggesting that non-coding RNA profiles obtained on ICU admission may not be suitable as mortality predictors.

## Protein-coding and non-coding RNA profiles of sepsis patients relative to human endotoxemia

Previous studies have compared the protein-coding RNA response in patients with sepsis or trauma (non-septic) to the response after lipopolysaccharide (LPS) administration to healthy volunteers in a controlled clinical setting (human endotoxemia) (*Cheng et al., 2016*; *Calvano et al., 2005*; *Scicluna et al., 2013*; *Perlee et al., 2018*; *Seok et al., 2013*; *Xiao et al., 2011*; *Takao and Miyakawa, 2015*). Here we sought to extend on those observations by evaluating long and small non-coding RNA expression in sepsis relative to temporal leukocyte responses in human endotoxemia (*Figure 3*). As previously reported in this model (*Cheng et al., 2016*; *Calvano et al., 2005*; *Scicluna et al., 2013*; *Perlee et al., 2018*), robust alterations in protein-coding RNA expression were noted after 2, 4, and 6 hr of LPS administration (*Figure 3—figure supplement 1*). Fold expression in sepsis (relative to health) was directly correlated with fold expression after 2, 4, and 6 hr LPS (*Figure 3A*). Long non-coding RNA expression was robustly altered in endotoxemia, with 2361, 5053, 2925, and 43 significant differences after 2, 4, 6, and 24 hr endotoxemia, respectively (*Figure 3—figure supplement 2A*). Pseudogenes, lincRNA, and antisense RNA were the most abundant long non-coding RNA biotypes (*Figure 3B*). Small non-coding RNA were modestly altered in human endotoxemia (*Figure 3—figure supplement 2B*). The most abundant biotypes of small RNA were miRNA (*Figure 3C*). Comparing fold expression in sepsis (relative to health) to human endotoxemia revealed significant correlations after 2, 4, and 6 hr of endotoxemia (*Figure 3D*). The highest $r^2$ was found for sepsis and 4 hr post-LPS ($r^2$ = 0.51). Correlation analysis of small RNA fold expression during endotoxemia against fold expression in sepsis revealed indirect correlations (*Figure 3E*).

## Functional inference of non-coding RNA

To better understand the functional organization of the non-coding leukocyte transcriptome in sepsis, particularly long non-coding RNA, we undertook a guilt-by-association approach. On the basis of a bi-weight midcorrelation matrix of the most variable protein-coding and long non-coding RNA

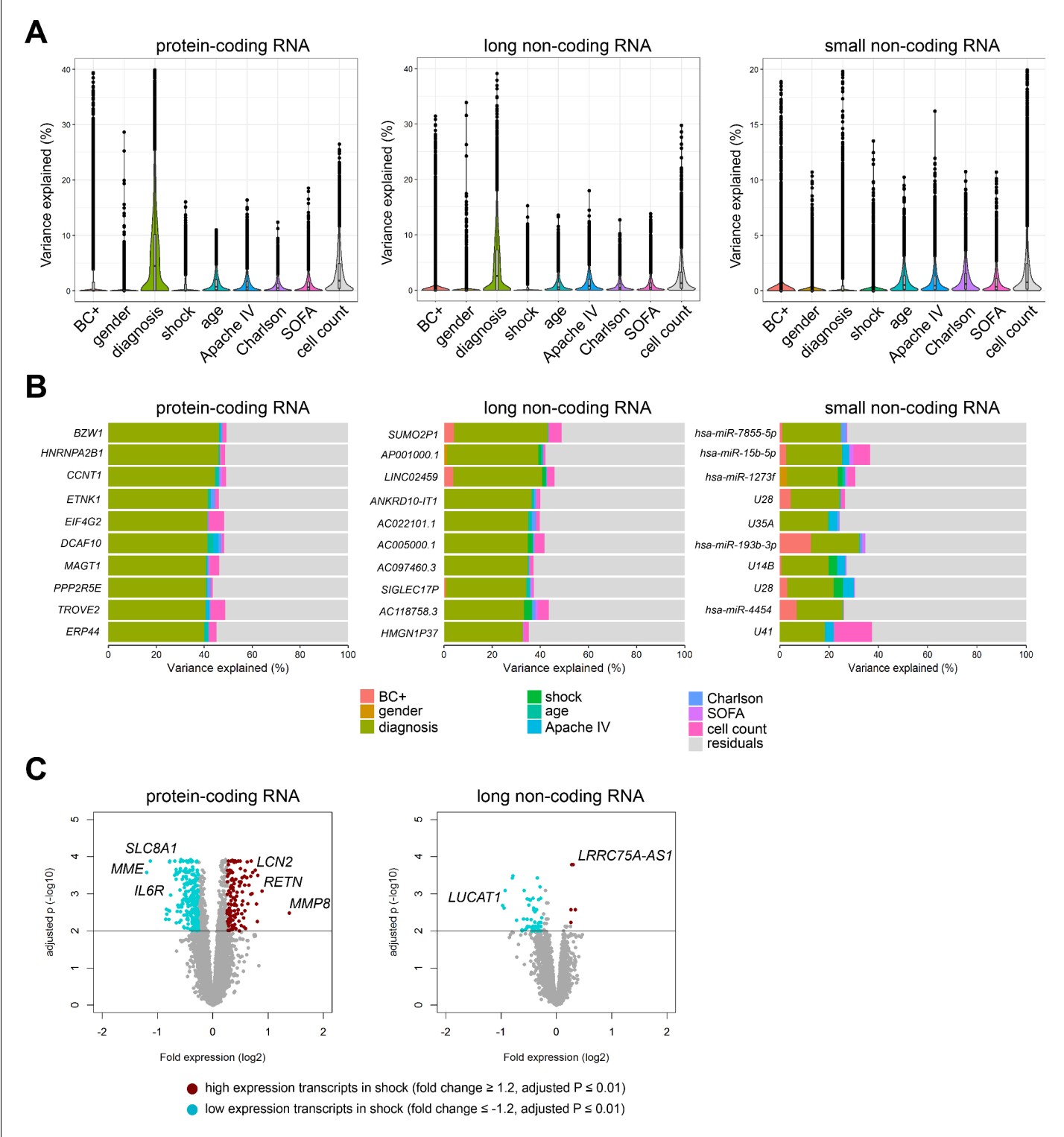

**Figure 2.** Variance in coding and non-coding RNA expression attributed to demographics and clinical characteristics of sepsis patients. (**A**) Violin plots of percent variation in protein-coding, long non-coding, and small non-coding RNA expression explained by sepsis patient demographics and clinical variables. Black dots depict outlier RNA transcripts. (**B**) Percent variance of select protein-coding and long non-coding RNA partitioned into the segment attributable to each demographic and clinical variable ranked by percent variation (>20%) for primary diagnosis (site of infection and place of acquisition). (**C**) Volcano plots depicting the changes in protein-coding and long non-coding RNA in patients discordant for septic shock on ICU admission. Horizontal (black) line denotes the adjusted p-value threshold for significance (adjusted p≤0.01). Abbreviations: BC+, blood culture positive

*Figure 2 continued on next page*

*Figure 2 continued*

microbiology; diagnosis, infection site (lung or abdomen) and source (community or hospital); Charlson, Charlson comorbidity index; Apache IV, Acute Physiology and Chronic Health Evaluation; ICU, Intensive care unit; SOFA, Sequential Organ Failure Assessment.
The online version of this article includes the following figure supplement(s) for figure 2:

**Figure supplement 1.** Variance partition of protein-coding and non-coding RNA expression in health and differential expression in sepsis non-survivors relative to survivors.

(n = 8539; coefficient of variation >5%) in sepsis patients only (*Figure 4*), a weighted network was built with scale-free topology (*Figure 4—figure supplement 1A*; *Langfelder and Horvath, 2008*; *Langfelder and Horvath, 2012*; *Scicluna et al., 2015a*). Hierarchical clustering uncovered 23 network modules (clusters) each harboring more than 100 inter-correlating RNA transcripts (*Figure 4A* and *Figure 4—figure supplement 1B*). Of the 8539 RNA transcripts, 158 transcripts did not cluster (designated as a gray module). Seventeen modules were associated with specific gene ontologies or canonical signaling pathways that included cell death/olfactory receptor activity/cell-cycle G2/M DNA damage checkpoint and regulation (turquoise module, n = 1001 transcripts) and RNA biosynthesis/RNA binding (yellow module, n = 579 transcripts) (*Figure 4A*). Eight modules in the co-expression network were significantly enriched for long non-coding RNA relative to protein-coding RNA (Fisher's adjusted p<0.01; *Figure 4B*). This suggests that the leukocyte long non-coding transcriptome of sepsis patients is primarily co-expressed with protein-coding RNA, but 34% of non-coding RNA modules were organized into distinct units. Evaluation of total and intra-module connectivities, which measure the importance of each module relative to the overall structure of co-expression networks (*Langfelder and Horvath, 2008*), identified two 'driver' modules, namely the cell death/olfactory receptor activity/cell-cycle G2/M DNA damage checkpoint and regulation (turquoise module, n = 1001 transcripts) and RNA biosynthesis/RNA binding (yellow module, n = 579 transcripts) modules (*Figure 4C and D* and *Figure 4—figure supplement 1C*). The former module included protein-coding RNA encoding ATM serine/threonine kinase (*ATM*), TNF alpha-induced protein 3 (*TNFAIP3* or A20), histone deacetylase 2 (*HDAC2*), and mucosa-associated lymphoid tissue lymphoma translocation protein 1 (*MALT1*) paracaspase (*Figure 4D*). Non-coding RNA included *GABPB1-AS1*, *THAP9-AS1*, and *SCARNA9*. We subsequently focused our attention on integrating miRNA profiles to the co-expression network. Considering miRNA profiles that were significantly altered in sepsis patients relative to health (*Figure 1C*), and miRNA-to-gene interactions (miRWalk method), we detected 49 small RNAs in five network modules with explained variance estimated >20%, including *hsa-miR-200c-3p* (translation initiation module), *SNORD84* (regulation of cytokine secretion/Toll-like receptor [TLR] signaling module), *HBII-276* (translation initiation module), *hsa-miR-1275* (sensory perception of chemical stimulus/olfactory receptor activity module), and *hsa-miR-664b-3p* (neutrophil degranulation/extracellular exosome module) (*Figure 4E*). Of note, *hsa-miR-200c-3p* has been shown to modify TLR4 signaling efficiency dependent on MYD88-mediated pathways in an embryonic kidney cell line (HEK293) (*Wendlandt et al., 2012*).

Next, we evaluated the association of network modules with soluble mediators of the host response and clinical severity scores. Neutrophil degranulation (secretory; red), protein ubiquitination (pink), and mitotic cell cycle (tan) modules correlated with soluble mediators of inflammation (C reactive protein [CRP], IL-6, IL-10, IL-8), endothelial responses (E-selectin and angiopoietin-2 [ANG2]), coagulation (D-Dimer), and clinical variables of disease severity (*Figure 5A*). In contrast, antigen presentation/Th1-Th2 cell activation (green module), regulation of cytokine secretion/TLR signaling (black module), and type-I interferon signaling/double stranded RNA binding (salmon module) were indirectly correlated with various soluble mediators and clinical severity indices. Patients with septic shock showed significantly higher neutrophil degranulation (secretory) expression patterns (*Figure 5B*). Protein-coding RNA transcripts in the neutrophil degranulation (secretory) module included matrix metalloproteinases (*MMP8* and *MMP9*), neutrophil activation cluster of differentiation 177 (*CD177*), lipocalin 2 (*LCN2*), and arginase 1 (*ARG1*) (*Figure 5C*). LincRNA and antisense RNA included an inducer of differentiation *MYOSLID* (myocardin-induced smooth muscle LncRNA, inducer of differentiation), cell proliferation, and metastasis-associated antisense RNA of the titin gene (*TTN-AS1*) and an IL10 receptor beta subunit antisense RNA, *IL10RB-AS1*. Calculating intra-modular connectivities enabled us to define 'hub' transcripts, which are understood to represent

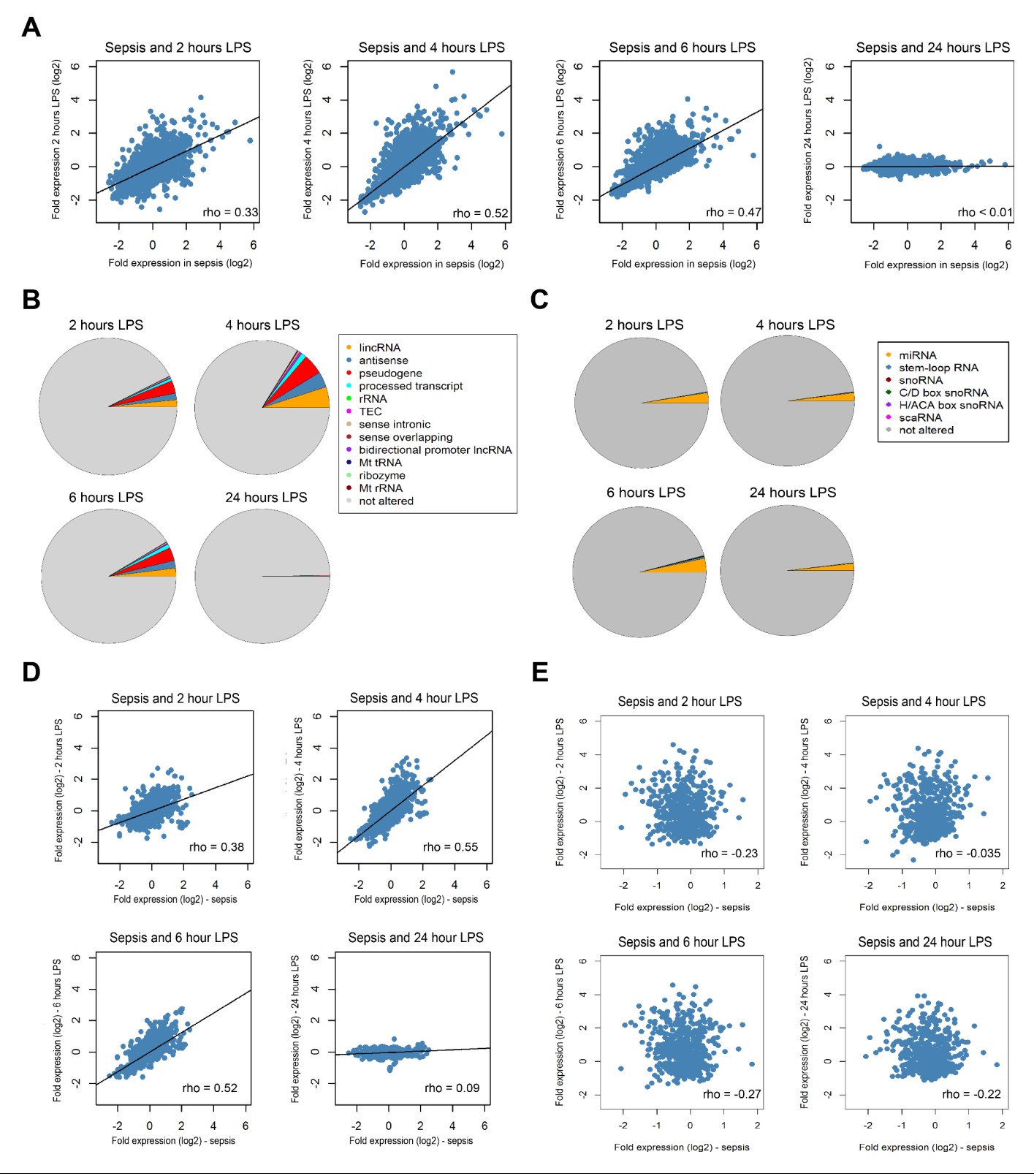

**Figure 3.** Comparison of the coding and non-coding transcriptome in sepsis to human endotoxemia. (**A**) Dot plots depicting the correlation between protein-coding RNA fold expression indices in sepsis (compared to health) and fold expression after 2, 4, 6, and 24 hr lipopolysaccharide (LPS) infusion relative to pre-LPS. (**B**) Pie chart illustrating the biotypes of significantly altered long non-coding RNA (adjusted p<0.01) across endotoxemia time points (2, 4, 6, and 24 hr after 2 ng/kg LPS). LincRNA, long intergenic non-coding RNA; rRNA, ribosomal RNA; TEC, to be experimentally confirmed; Mt tRNA,

*Figure 3 continued on next page*

*Figure 3 continued*

mitochondrial transfer RNA; Mt rRNA, mitochondrial ribosomal RNA. (C) Pie chart showing the biotypes of significantly altered small non-coding RNA (adjusted p<0.05) in human endotoxemia. miRNA, microRNA; snoRNA, small nucleolar RNA; C/D box snoRNA, C/D box small nucleolar RNA; H/ACA box snoRNA, H/ACA box small nucleolar RNA; scaRNA, small cajal body-specific RNA. (D) Dot plots illustrating the correlation between long non-coding RNA fold expression indices in sepsis (compared to health) and fold expression of 2, 4, 6, and 24 hr after LPS relative to pre-LPS. rho, Spearman's coefficient. (E) Dot plots depicting the correlation between small non-coding RNA fold expression indices in sepsis (compared to health) and 2, 4, 6, and 24 hr after LPS relative to pre-LPS. rho, Spearman's coefficient.

The online version of this article includes the following figure supplement(s) for figure 3:

**Figure supplement 1.** Comparing fold expression in sepsis (relative to health) to human endotoxemia.

**Figure supplement 2.** Volcano plot representations of significantly altered.

cogs in the functional output of a network module (*Langfelder and Horvath, 2008*; *Zhao et al., 2010*), and identified *MYOSLID* (neutrophil degranulation; red module) and *LUCAT1* (Lung Cancer Associated Transcript 1) in the TLR-signaling (black) module, as module 'hubs'. In line with their respective module eigengene correlations with inflammatory response markers, *MYOSLID* expression was directly correlated with levels of inflammatory response markers IL-6, IL-8, IL-10, and acute phase response protein CRP (*Figure 5D*). In contrast, *LUCAT1* expression was indirectly correlated with soluble mediators of inflammation, except for CRP (*Figure 5E*).

## Discussion

In this study we found that the transcriptional changes in critically ill patients with sepsis are not exclusive to protein-coding RNAs. Whole blood long non-coding RNAs, and to a lesser extent small non-coding RNAs, were significantly altered in sepsis patients relative to healthy subjects. The pattern of protein-coding and long non-coding RNA profiles in sepsis was mimicked by expression profiles in a human endotoxemia model, notably at a time point indicative of endotoxin tolerance. Small non-coding RNA profiles in sepsis patients were not recapitulated in human endotoxemia. In general, common clinical characteristics explained low proportions of variation in protein-coding and non-coding RNA profiles, suggesting that variation in leukocyte responses are largely not explained by clinical parameters. Leveraging on the concepts of network biology, protein-coding and non-coding RNA were clustered as functional biological units with RNA binding/RNA biosynthesis and cell death/olfactory receptor activity/cell-cycle G2-M DNA damage checkpoint and regulation modules central to network architecture.

Advances in genomics, notably massively parallel cDNA sequencing, have shown that active transcription is not exclusive to protein-coding RNA regions (*Carninci et al., 2005*). Regions of the genome void of protein-coding genes have since been shown to be actively transcribed in the context of various diseases (*Esteller, 2011*). Small non-coding RNAs, mainly microRNAs, as well as long non-coding RNAs were linked to specific immune processes (*Mehta and Baltimore, 2016*; *Fitzgerald and Caffrey, 2014*). While microRNAs have been established as veritable epigenetic modifiers of transcriptional outputs, studies on the functional aspects of long non-coding RNAs have only recently begun. However, those studies were centered primarily on mouse models (*Atianand et al., 2016*; *Carpenter et al., 2013*). This presents a problem for translation to human physiology because non-coding RNA sequences are typically not conserved between species (*Diederichs, 2014*). Furthermore, expression of non-coding RNAs was shown to exhibit substantially higher inter-individual variation in healthy subjects as compared to protein-coding RNAs alone (*Kornienko et al., 2016*). In line with those observations our data showed that long non-coding RNA expression patterns were far more variable across individuals (healthy or sepsis) than protein-coding and small non-coding RNAs. The sources of increased inter-individual variation in long non-coding RNA expression relative to protein-coding and small non-coding RNAs are as yet unknown. Lower conservation coupled with faster evolution rates of long non-coding RNA regions, which seemingly harbor more single nucleotide polymorphisms (SNPs) than protein-coding genes (*Necsulea and Kaessmann, 2014*), as well as the possibility of their relatively higher susceptibility to environmental and lifestyle factors (*Dumeaux et al., 2010*), may be at the basis of the extensive variation in long non-coding RNA expression.

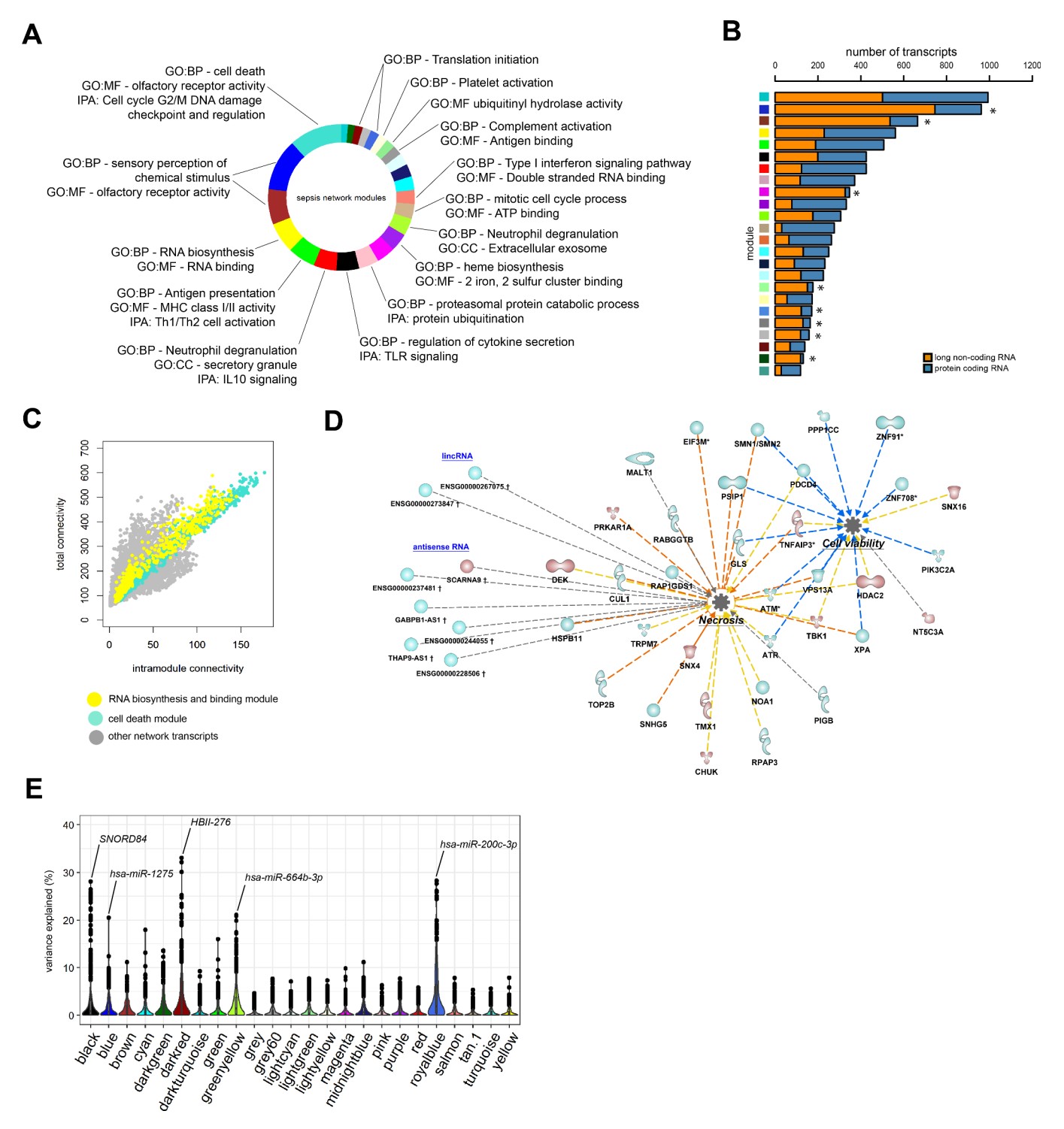

**Figure 4.** Network analysis of coding and non-coding RNA expression. (**A**) Circular plot of protein-coding and long non-coding co-expression network modules characterized by significantly associated (Fisher's adjusted p<0.01) gene ontologies and Ingenuity canonical signaling pathways. Seventeen modules were associated with specific ontologies or canonical signaling pathways. (**B**) Bar plot depicting the distribution of protein coding and long non-coding RNA in each network module. *Fisher's Benjamini–Hochberg adjusted p<0.01. (**C**) Dot plot illustrating the correlation between intramodular and total connectivities of each RNA transcript in their respective network module. Yellow dots illustrate protein-coding and long non-coding RNA in the RNA biosynthesis/RNA binding module; Turquoise dots depict the cell death and olfactory receptor activity module. (**D**) Diagrammatic representation of Ingenuity's biofunctions (z-score <2 or >2 and adjusted p<0.05) together with predicted long intergenic non-coding RNA (lincRNA)

*Figure 4 continued on next page*

*Figure 4 continued*

and antisense RNA in the cell death/olfactory receptor activity/cell-cycle G2/M DNA damage checkpoint and regulation module (turquoise). Blue, reduced expression; red, elevated expression in sepsis relative to health (fold change $\geq$1.2 or $\leq-$1.2; adjusted p-value<0.01). (E) Violin plots of network module eigengene (first principal component) percent variance attributable to small non-coding RNA.

The online version of this article includes the following figure supplement(s) for figure 4:

**Figure supplement 1.** Co-expression network analysis.

In line with previous studies (*Seok et al., 2013*; *Takao and Miyakawa, 2015*), we found that protein-coding RNA alterations during endotoxemia mimicked those that ensue in sepsis patients. The human endotoxemia model is a highly relevant in vivo model of acute systemic inflammation in the context of a controlled clinical setting (*Lowry, 2005*). In general, the model is characterized by a robust systemic response, including leukocyte transcriptional responses, exhibiting shared and unique temporal changes that resolve within 24 hr of bolus administration (*Calvano et al., 2005*; *Perlee et al., 2018*). In extension to the previously reported data, based on a single time-point of human endotoxemia (*Seok et al., 2013*; *Takao and Miyakawa, 2015*), we found that the correlation between sepsis and human endotoxemia was also dependent, at least in part, on timing of the response to LPS. The highest correlation was found at 4 hr, a time point at which the capacity of cytokine production by leukocytes is typically reduced in the human endotoxemia model, indicative of endotoxin tolerance (*Cheng et al., 2016*; *de Vos et al., 2009*). Long non-coding RNA alterations in human endotoxemia also mimicked those in sepsis, with similar time dependencies as protein-coding RNA. In contrast, small non-coding RNA expression profiles in sepsis patients were not reliably recapitulated in human endotoxemia, primarily showing indirect correlations. This may be due to typically low expression patterns of miRNA, compared to protein-coding and long non-coding RNA, and reported high specificities of miRNA to developmental stage and cell-type (*Bernstein et al., 2003*). The host response during infection is characterized by a balance between resistance (seeking to limit the pathogen load) and tolerance (aiming to retain cell and organ functions) (*Schneider and Ayres, 2008*). In sepsis both mechanisms can become uncontrolled, wherein aberrant activation of resistance pathways results in tissue damage and inadequate tolerance can cause immune suppression with enhanced susceptibility to secondary infections (*Bauer and Wetzker, 2020*). While our time-sequential data in healthy humans injected with LPS suggest that coding and long non-coding RNA profiles in blood leukocytes of sepsis patients particularly reflect a tolerant state, time course studies in patients are needed to increase the insight into the role of distinct RNA species in the interplay between resistance and tolerance.

A substantial proportion of variance in protein-coding and non-coding RNA expression in critically ill patients with sepsis remained unexplained. Other sources of variation, not assessed in this study, include patient genetics and time between the onset of sepsis and ICU admission (*Schadt et al., 2003*; *Maslove and Wong, 2014*). The former represents an important source of inter-individual variation where SNPs segregating in populations are in part tightly related to RNA expression variability (*Schadt et al., 2003*). This was shown in a recent prospective study in sepsis due to community-acquired pneumonia (CAP), wherein SNPs influencing gene expression patterns were identified (*Davenport et al., 2016*). The time of onset of sepsis is a current 'black box' in the field as it cannot be accurately determined, thereby resulting in considerable uncertainty since patients are presumably admitted to the ICU at various stages of the sepsis syndrome. Overall, we determined that clinical characteristics and outcome explained low proportions of variation in RNA expression; however, specific protein-coding and long non-coding RNA transcripts had high percent variation attributable to, particularly, primary diagnosis that included infections site (lung or abdomen) and place of acquisition (community or hospital), which may constitute important proxies to discern organ-specific infections that are typically caused by different causal pathogens (*van Vught et al., 2016a*; *van Vught et al., 2016b*; *Sartelli, 2010*).

Ascribing long non-coding RNA function to cellular biological pathways is a major challenge. To address this challenge, we undertook a guilt-by-association strategy that sought to position long non-coding RNA in co-expression modules of tightly correlating protein-coding RNA, thereby infer on functional outputs of long non-coding RNA by virtue of the pathways that associate with protein-coding RNA in each module. By leveraging on the concepts of scale free networks (*Barabási, 2009*),

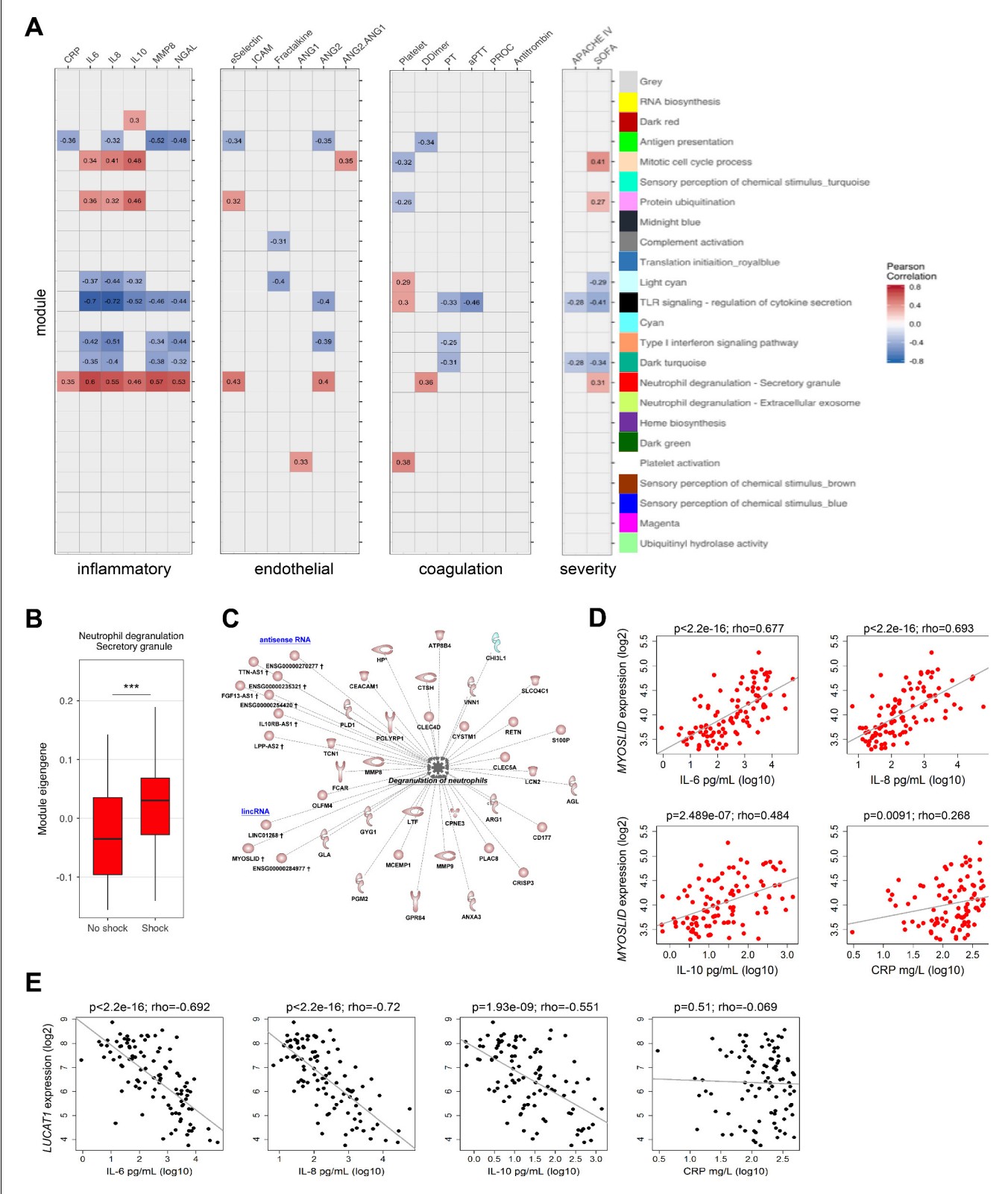

**Figure 5.** Relationship of protein-coding, non-coding RNA network modules to soluble mediators and clinical severity. (**A**) Heatmap representation of Pearson correlation coefficients (adjusted p<0.05) calculated for each network module eigengene (first principal component) against soluble mediators of inflammation, endothelial function, coagulation, as well as clinical parameters of disease severity. APACHE IV, Acute Physiology and Chronic Health Evaluation; SOFA, Sequential Organ Failure Assessment. Red denotes direct correlations and blue denotes indirect correlations (**B**) Boxplot showing

*Figure 5 continued on next page*

*Figure 5 continued*

differences in neutrophil degranulation (red) module eigengene values in sepsis patients discordant for septic shock on intensive care unit admission. High module eigengene values mean overall elevated RNA expression; low module eigengene values mean reduced expression. (C) Diagrammatic representation of the neutrophil degranulation (secretory; red) module (Ingenuity's biofunction z-score <2 or >2; adjusted p<0.05) together with predicted long intergenic non-coding RNA (lincRNA) and antisense RNA. Red or blue nodes denote high or low expression RNA transcripts in sepsis relative to health, respectively. ***Mann–Whitney p<0.001. (D and E) Dot plots of (D) *MYOSLID* expression and (E) *LUCAT1* expression against soluble mediators of inflammation IL-6, IL-8, and IL-10, as well as the acute phase response protein CRP. Rho, Spearman's coefficient.

we built a map of protein-coding and non-coding RNA relationships that pointed to cell death/olfactory receptor activity/cell-cycle G2/M DNA damage checkpoint and regulation (turquoise module) and RNA biosynthesis/RNA binding (yellow module) as central to the organization of the co-expression network. Cell death or exhaustion, particularly in lymphocytes, have been proposed as causal features of immunosuppression and lethality in sepsis (*Hotchkiss et al., 2013*). Our findings further strengthen this hypothesis and position previously unknown non-coding RNA, including an autophagy and chemical stress responder *GABPB1-AS1* (*Tani et al., 2014*; *Luan et al., 2019*), as putative regulators of cell death in the context of sepsis. Interestingly, protein-coding RNA in the cell death (turquoise) module also included olfactory receptors and cell-cycle DNA damage regulators. Modulation of DNA damage responses was demonstrated as a potential therapeutic path that might be exploited to confer protection to severe sepsis (*Figueiredo et al., 2013*). Little is known about olfactory receptors in non-chemosensory cells, but a growing body of evidence suggests they are not exclusive to the nose (*Kang and Koo, 2012*). They have been shown to be involved in cell–cell recognition, migration, proliferation, and apoptosis (*Maßberg and Hatt, 2018*).

In conclusion, we here describe the non-coding RNA landscape in blood leukocytes of sepsis patients upon admission to the ICU. By considering non-coding RNA expression patterns in relation to protein-coding RNA we provide an important layer to the blood leukocyte 'regulome' in a clinical context, which may facilitate prioritization of non-coding RNA in future functional studies.

# Materials and methods

**Key resources table**

| Reagent type (species) or resource | Designation | Source or reference | Identifiers | Additional information |
|---|---|---|---|---|
| Biological sample (*Homo sapiens*) | Total RNA | Leukocytes | | |
| Commercial assay or kit | PAXgene Blood miRNA kit | Qiagen | Cat no./ID: 763134 | |
| Commercial assay or kit | Human Transcriptome Array 2.0 | Affymetrix; Thermo Fisher | | microarray |
| Commercial assay or kit | miRNA 4.1 96-array plate | Affymetrix; Thermo Fisher | | microarray |
| Commercial assay or kit | FlexSet cytometric bead arrays | BD Biosciences | | |
| Commercial assay or kit | Immuno turbidimetric assay | Roche diagnostics | | |
| Commercial assay or kit | Luminex Flow Cytometry Analyzer | Luminex Corp. | RRID:SCR_018025 | |
| Commercial assay or kit | Sysmex CA-1500 System | Siemens Healthineers | | |

*Continued on next page*

*Continued*

| Reagent type (species) or resource | Designation | Source or reference | Identifiers | Additional information |
|---|---|---|---|---|
| Chemical compound, drug | Lipopolysaccharide-*Escherichia coli*, 100 ng/ml, Ultrapure | Invivogen | Cat#0111:B4 | |
| Software, algorithm | R Project for Statistical Computing, (version 3.5.0) | R Development Core Team | RRID:SCR_001905 | |
| Software, algorithm | Oligo (version 1.44) | Bioconductor (*Carvalho and Irizarry, 2010*) | RRID:SCR_015729 | |
| Software, algorithm | SVA (version 3.28) | Bioconductor (*Leek and Storey, 2007*) | RRID:SCR_012836 | |
| Software, algorithm | genefilter (version 1.62) | Bioconductor (*Bourgon et al., 2010*) | | |
| Software, algorithm | arrayQuality Metrics | Bioconductor (*Kauffmann et al., 2009*) | RRID:SCR_001335 | |
| Software, algorithm | Affymetrix Transcriptome Analysis Console | Affymetrix | RRID:SCR_018718 | |
| Software, algorithm | limma (version 3.36) | Bioconductor (*Smyth, 2005*) | RRID:SCR_010943 | |
| Software, algorithm | Ingenuity pathway analysis software | Qiagen | RRID:SCR_008653 | |
| Software, algorithm | WGCNA (version 1.64) | Bioconductor (*Langfelder and Horvath, 2008*) | RRID:SCR_003302 | |
| Software, algorithm | miR-Walk 2.0 | University of Heidelberg, Germany (*Dweep et al., 2011*) | | |
| Software, algorithm | variance Partition (version 1.10) | Bioconductor (*Hoffman and Schadt, 2016*) | | |
| Software, algorithm | mixOmics | Bioconductor (*Rohart et al., 2017*) | RRID:SCR_016889 | |
| Other | Deposited data super-series | Gene Expression Omnibus | GSE134364 | |

## Patient population and inclusion criteria

This study was part of the Molecular Diagnosis and Risk Stratification of sepsis (MARS) project, a prospective observational study in the mixed ICUs of two tertiary teaching hospitals in the Netherlands (Academic Medical Center, Amsterdam and University Medical Center Utrecht, Utrecht) (ClinicalTrials.gov identifier NCT01905033) (*van Vught et al., 2016a*; *Klein Klouwenberg et al., 2013*; *Scicluna et al., 2015b*). For the current study, we selected consecutive patients with sepsis from the MARS biorepository who were older than 18 years of age and had been admitted to the ICU between July 2012 and January 2014. Sepsis (n = 156) was defined as the presence of CAP, hospital-acquired pneumonia (HAP), or intra-abdominal infection diagnosed within 24 hr of ICU admission with a culture proven or probable likelihood using criteria as described (*Zimmerman et al., 2006*), accompanied by at least one additional general, inflammatory, hemodynamic, organ dysfunction, or tissue perfusion variable described in the third international consensus definitions for sepsis and septic shock (*Singer et al., 2016*). Patients with aspiration pneumonia, with multiple sites of infection, and patients admitted to the ICU more than 2 days after the initiation of antibiotics were excluded. All readmissions and patients transferred from another ICU were also excluded, except

when patients were referred to one of the study centers on the same day of presentation to the first ICU. Severity was assessed by APACHE IV (*Zimmerman et al., 2006*) and SOFA score excluding the central nervous system component (*Vincent et al., 1996*). Shock was qualified by the use of vasopressors (norepinephrine, epinephrine, or dopamine) for hypotension in a norepinephrine-equivalent dose of more than 0.1 µg/kg/min in patients with a SOFA score of at least 2 (*Singer et al., 2016*). Blood was collected in PAXgene tubes (Becton-Dickinson, Breda, The Netherlands) and ethylenediaminetetraacetic acid (EDTA) vacutainer tubes within 24 hr of ICU admission. Definitions of comorbid and immunocompromised conditions are reported in the online data supplement.

## Healthy participants and endotoxemia

PAXgene and EDTA tubes were also obtained from 82 healthy subjects. Eight male subjects were exposed to intravenous LPS in a Phase I, randomized, single-blind, parallel group, placebo controlled study (Clinicaltrials.gov identifier NCT02328612); the subjects who received placebo were used in the current study (*Perlee et al., 2018*). Subjects were infused with LPS over 1 min (2 ng/kg; from Escherichia [*E. coli*], US standard reference endotoxin, kindly provided by Anthony Suffredini, National Institute of Health, Bethesda, MD). Whole blood was collected in PaxGene Blood tubes (Qiagen) before and 2, 4, 6, and 24 hr after LPS administration.

## Immunological markers

EDTA-anticoagulated blood plasma collected on ICU admission was used for soluble mediator measurements. IL-6, IL-8, IL-10, soluble intercellular adhesion molecule-1 (ICAM-1), soluble E-selectin, and fractalkine were measured using FlexSet cytometric bead arrays (BD Biosciences, San Jose, CA) using a FACS Calibur (Becton Dickinson, Franklin Lakes, NJ, NJ, USA). Neutrophil gelatinase-associated lipocalin (NGAL), ANG-1, ANG-2, protein C, antithrombin, matrix metalloproteinase (MMP)−8 (R and D Systems, Abingdon, UK), and D-dimer (Procartaplex, eBioscience, San Diego, CA) were measured by Luminex multiplex assay using a BioPlex 200 (BioRas, Hercules, CA). CRP was determined by an immunoturbidimetric assay (Roche diagnostics). Platelet counts were determined by hemocytometry, prothrombin time (PT), and activated partial thromboplastin time (aPTT) by using a photometric method with Dade Innovin Reagent or by Dade Actin FS Activated PTT Reagent, respectively (Siemens Healthcare Diagnostics). Normal biomarker values were obtained from 27 age- and sex-matched healthy subjects, except for CRP, platelet counts, PT, and aPTT (routine laboratory reference values).

## Microarrays and data processing

Total RNA was isolated by means of PaxGene blood miRNA isolation kit (Thermo-Fisher) as per manufacturer's instructions. Quality RNA (Agilent 2100 Bioanalyzer, Agilent Technologies; RIN > 6) was processed and hybridized to either the GeneChip Human Transcriptome Array (HTA) 2.0 (Thermo-Fisher) or the miRNA 4.1 96-array plate (Thermo-Fisher) following manufacturer's instructions. Both arrays were done on all samples (sepsis patients, controls, and healthy subjects injected with LPS). Microarrays were scanned at the Cologne Center for Genomics, Cologne, Germany.

The HTA 2.0 scans (.CEL) were processed in the R language and environment for statistical computing version 3.5.0 (R Development Core Team, Foundation for Statistical Computing, Vienna, Austria). Following robust multi-average (RMA) background-correction, quantile normalization, and $\log_2$-transformation using the oligo method (version 1.44) (*Carvalho and Irizarry, 2010*) data were evaluated for non-experimental chip effects by means of surrogate variable analysis (SVA; version 3.28) and adjusted using the combat method (*Leek and Storey, 2007*). Probes were annotated using biomart (version 2.36.1) (*Smedley et al., 2015*), and low expression probes were filtered by means of the genefilter method (version 1.62) (*Bourgon et al., 2010*). The miRNA-4.1 scans (.CEL) were analyzed by means of Affymetrix Expression Console software (Thermo-Fisher). Probes were normalized using the RMA method and detection above background (DABG) probe level detection. *Homo sapiens* annotated probes with detection p-value <0.05 in at least one sample were considered for downstream analyses. Quality of HTA2.0 and miRNA-4.1 arrays was evaluated by means of the array-qualitymetrics R package (*Kauffmann et al., 2009*). Comparisons between study groups were done using the limma method (version 3.36) (*Smyth, 2005*) and significance was demarcated by Benjamini–Hochberg multiple test adjusted probabilities (adjusted p<0.01). The linear model included age

and sex as additive covariates. The MDTH index was calculated as described previously (*Berry et al., 2010*; *Dunning et al., 2018*). Ingenuity Pathway Analysis (Ingenuity systems, Qiagen) was used to determine the most significant canonical signaling pathways for elevated and reduced protein-coding RNA transcripts considering adjusted Fisher's probabilities (adjusted p<0.05) specifying the Ingenuity knowledgebase as reference and human species. All other parameters were default.

The novelty of our study, that is, profiling non-coding RNA expression in leukocytes of patients with sepsis, precludes an adequate study power estimation. However, considering known co-regulation with protein-coding RNA expression, we provide study power estimates based on previous observations in typical gene expression studies (*Cheng et al., 2016*; *Scicluna et al., 2017*; *Davenport et al., 2016*). Considering a false discovery rate of 5%, beta error level 5% (95% power), and typical effect sizes greater than 0.25 in sepsis relative to health, a sample size of 42 per group was estimated. In addition, eight healthy volunteers in a human endotoxemia challenge would have more than 95% power to detect differences relative to pre-challenge (baseline) samples (*Cheng et al., 2016*; *Davenport et al., 2016*; *Calvano et al., 2005*; *Scicluna et al., 2013*; *Perlee et al., 2018*; *Seok et al., 2013*; *Xiao et al., 2011*; *Takao and Miyakawa, 2015*). Using a continuous model, we estimated that 156 patients would have more than 98% power to detect significant associations with demographic or clinical variables (false-discovery rates of 5%).

## Co-expression network and pathway analysis

The weighted gene co-expression network analysis (WGCNA) method (version 1.64) was used to build the leukocyte co-expression network as described previously (*Langfelder and Horvath, 2008*; *Scicluna et al., 2015a*; *Zhao et al., 2010*). A pair-wise biweight midcorrelation matrix of the most variable transcripts (coefficient of variation >5%) was transformed into an adjacency matrix by using a 'soft' power function of 8 ensuring scale-free topology (*Langfelder and Horvath, 2008*; *Zhao et al., 2010*). The adjacency matrix was further transformed into a topological overlap matrix to enable the identification of modules (clusters) encompassing highly inter-correlating RNA transcripts by using a dynamic tree cut method (version 1.63) (*Langfelder and Horvath, 2008*; *Zhao et al., 2010*). Modules were summarized by means of the eigengene value, defined as the first PC of the module expression matrix and the module membership measure. Protein-coding RNA in each module were analyzed for enrichment of gene ontologies for biological processes (GO:BP), molecular function (GO:MF), and cellular compartment (GO:CC) using the Gene Ontology Consortium database with significance defined by adjusted p-value <0.05 (http://www.geneontology. org) (*Ashburner et al., 2000*). Biofunctions were predicted using Ingenuity Pathways software (Ingenuity pathway analysis, Qiagen Bioinformatics) specifying activation z-score <2 or >2 and adjusted p-value <0.05. The miR-Walk atlas of gene-miRNA-target interactions was used to evaluate predicted interactions of miRNA with module-specific genes by specifying the miR-Walk algorithm (*Dweep et al., 2011*; *Mills et al., 2017*). Human species annotations and 3' untranslated region (UTR) interactions as well as a minimum seed length equating to seven were specified. All other parameters were default.

## Statistics

Statistical analysis was performed in the R statistical environment (v 3.5.0). Comparison of continuous data between categories was done with the Wilcoxon rank sum test. Correlation analysis of continuous data was performed using Pearson's method unless otherwise stated as well as the coefficient of determination ($r^2$). Categorical data were analyzed by Fisher exact tests or chi-squared tests. Multiple comparison (Benjamini–Hochberg) adjusted p-values <0.05 defined significance. The proportion of variance in RNA expression explained by demographics and clinical characteristics was calculated using a multivariate approach implemented in the variancePartition method (version 1.10) (*Hoffman and Schadt, 2016*). A multivariate linear model was fit including age, gender, primary diagnosis, total SOFA, APACHE IV scores, shock, and Charlson comorbidity indices. PC analysis was done using the mixOmics package, specifying 10 components (*Rohart et al., 2017*). Data is presented in the form of volcano plots, pie charts, dot plots, bar charts, and circular and violin plots.

# Acknowledgements

The authors thank all the patients and healthy volunteers who participated in this study, as well as the critical care nursing staff at both the AMC and UMCU ICUs. Members of the MARS consortium were: from Amsterdam University Medical Centers, location Academic Medical Center, University of Amsterdam, the Netherlands: Friso M de Beer, Lieuwe D J Bos, Gerie J Glas, Roosmarijn T M van Hooijdonk, Janneke Horn, Mischa A Huson, Laura R A Schouten, Marleen Straat, Luuk Wieske, Maryse A Wiewel, and Esther Witteveen; from University Medical Center Utrecht, Utrecht, the Netherlands: David SY Ong, Jos F Frencken, Maria E Koster-Brouwer, Kirsten van de Groep, and Diana M Verboom.

# Additional information

## Funding

| Funder | Grant reference number | Author |
|---|---|---|
| Innovative Medicines Initiative Joint Undertaking | 115523 | Marc J Bonten |
| Innovative Medicines Initiative Joint Undertaking | 115620 | Marc J Bonten |
| Innovative Medicines Initiative Joint Undertaking | 115737 | Marc J Bonten |
| Center for Translational Molecular Medicine | 04I-201 | Tom van der Poll |

The funders had no role in study design, data collection and interpretation, or the decision to submit the work for publication.

## Author contributions

Brendon P Scicluna, Conceptualization, Data curation, Formal analysis, Investigation, Methodology, Writing - original draft; Fabrice Uhel, Formal analysis, Writing - review and editing; Lonneke A van Vught, Maryse A Wiewel, Arie J Hoogendijk, Olaf L Cremer, Marcus J Schultz, Resources, Writing - review and editing; Ingelore Baessman, Marek Franitza, Peter Nürnberg, Janneke Horn, Resources; Marc J Bonten, Resources, Funding acquisition, Writing - review and editing; Tom van der Poll, Conceptualization, Funding acquisition, Writing - original draft; Molecular Diagnosis and Risk Stratification in Sepsis (MARS) consortium, Project administration

## Author ORCIDs

Brendon P Scicluna (ID) https://orcid.org/0000-0003-2826-0341
Fabrice Uhel (ID) http://orcid.org/0000-0003-4946-8184

## Ethics

Human subjects: The institutional review boards of both participating centers approved an opt-out consent method (IRB No. 10-056C). The Dutch Central Committee on Research Involving Human Subjects and the Medical Ethics Committee of the Academic Medical Center, Amsterdam, the Netherlands, approved the study. Written informed consent was obtained from all healthy participants.

## Decision letter and Author response

Decision letter https://doi.org/10.7554/eLife.58597.sa1
Author response https://doi.org/10.7554/eLife.58597.sa2

# Additional files

## Supplementary files

• Supplementary file 1. Table of causative pathogens in critically ill patients with sepsis (n = 156). Percentages depict the proportion of infections caused by the pathogen indicated. In total, 192 pathogens were assigned to 156 infections. In 40 (25.6%) infections, more than one pathogen was assigned as causative.

• Transparent reporting form

## Data availability

The datasets generated and analyzed in the current study are available in the Gene Expression Omnibus of the National Center for Biotechnology Information repository with primary data accession numbers GSE134364 (super-series), GSE134347 for patients and healthy volunteers (HTA 2.0 microarray), GSE134356 for the human endotoxemia model samples (HTA 2.0 microarray) and GSE134358 for all patients, healthy volunteers and human endotoxemia samples (miRNA-4.1 microarray).

The following datasets were generated:

| Author(s) | Year | Dataset title | Dataset URL | Database and Identifier |
|---|---|---|---|---|
| Scicluna BP | 2020 | GEO super-series of patients and healthy volunteers | https://www.ncbi.nlm.nih.gov/geo/query/acc.cgi?acc=GSE134364 | NCBI Gene Expression Omnibus, GSE134364 |
| Scicluna BP | 2020 | HTA2.0 microarray data of patients and healthy volunteers | https://www.ncbi.nlm.nih.gov/geo/query/acc.cgi?acc=GSE134347 | NCBI Gene Expression Omnibus, GSE134347 |
| Scicluna BP | 2020 | HTA2.0 microarray data of human endotoxemia volunteers | https://www.ncbi.nlm.nih.gov/geo/query/acc.cgi?acc=GSE134356 | NCBI Gene Expression Omnibus, GSE134356 |
| Scicluna BP | 2020 | MicroRNA microarray data of patients, healthy volunteers and human endotoxemia volunteers | https://www.ncbi.nlm.nih.gov/geo/query/acc.cgi?acc=GSE134358 | NCBI Gene Expression Omnibus, GSE134358 |

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

# Appendix 1

## Patients

Comorbidities were defined as follows: cardiovascular compromise was defined as a medical history of congestive heart failure, chronic cardiovascular disease, myocardial infarction, peripheral vascular disease, or cerebrovascular disease. Malignancy was defined as a medical history of either metastatic or not metastatic solid tumor, or hemodynamic malignancy. Patients with a history of chronic renal insufficiency or treated with chronic intermittent hemodialysis or continuous ambulatory peritoneal dialysis were marked as renal insufficient. Respiratory insufficiency included patients with a history of chronic respiratory insufficiency, chronic obstructive pulmonary disease, or treated at home with oxygen or ventilator support. Patients with a history of immune deficiency, human immunodeficiency virus (HIV) infection, acquired immune deficiency syndrome (AIDS), asplenia, or chronically treated with corticosteroids, antineoplastic or other immune suppressive medications were deemed immunocompromised.

