## [Decision Letter]

**Acceptance summary:**

Sepsis is a serious clinical condition, in which an abnormal host response to infection leads to organ failure with a high risk of death. This study by Scicluna et al. assesses the role of non-coding RNA in patients with sepsis. Using advanced molecular analysis and bioinformatics the authors compare non-coding RNA patterns in sepsis patients and healthy subjects. In addition, the authors have treated healthy volunteers with bacterial endotoxin, which is considered an important factor in bacterial sepsis, to investigate the role non-coding RNAs. They have found that long non-coding RNAs, and to a lesser extent small non-coding RNAs, were significantly altered in sepsis patients as compared to healthy controls. Thereby, this paper significantly adds to the current concepts of the regulatory responses in sepsis.

**Decision letter after peer review:**

Thank you for submitting your article "The leukocyte non-coding RNA landscape in critically ill patients with sepsis" for consideration by *eLife*. Your article has been reviewed by three peer reviewers, including Evangelos J Giamarellos-Bourboulis as the Reviewing Editor and Reviewer #1, and the evaluation has been overseen by Jos van der Meer as the Senior Editor. The following individual involved in review of your submission has agreed to reveal their identity: Michael Bauer (Reviewer #3).

The reviewers have discussed the reviews with one another and the Senior Editor has drafted this decision to help you prepare a revised submission.

Consensus review proposal:

Non-coding RNA have been recently discovered as a new component of regulatory complexity of gene Expression in eukaryotic cells. The present study by Scicluna et al. is aimed to assess the role of non-coding RNA in sepsis patients. Using advanced molecular analysis and bioinformatics the authors compare non-coding RNA patterns in sepsis patients and healthy subjects. Additional, healthy volunteers have been treated with lipopolysaccharide to investigate effects of endotoxemia on non-coding RNAs. Resulting data are of novelty and significantly add to the current knowledge on sepsis related regulatory responses. The RNA preparation and analysis has been accomplished by up-to-date technology.

The paper is descriptive, well-written, and with reasonable conclusions. It is a reasonable step forward in the field, but to some degree the paper suffers from a lack of context. Other authors have written on this topic and no attempt is made to discuss, reproduce, or bioinformatically include/compare to those studies.

Essential revisions:

1) The authors might consider PMID: 29657277 (data on GEO). A very small study is here: https://doi.org/10.1089/gtmb.2017.0061. They also might consider mining their own dataset GSE65682 as others have done recently: https://translational-medicine.biomedcentral.com/articles/10.1186/s12967-020-02372-2. Many microarrays measure lncRNAs and the data are available, just not examined, online. Shouldn't the authors try to validate their findings externally?

2) As a brief aside, much of the above is also true for endotoxin studies. There are also a huge number of studies looking at miRNAs in sepsis in a mechanistic fashion; nice review here: PMC6255943. All of this could be contextualized better.

3) Why is the analysis of non-coding RNAs in this submission accompanied by an analysis of protein-coding regions? We understand that the authors try to investigate how the expression of coding regions is modulated by non-coding regions in terms of pathways analysis, but the way data are analysed does not support such conclusions.

4) Although there is a positive association of coding-RNAs between sepsis patients and the LPS endotoxemia model, this is at the best marginal for the non-coding RNAs. This should be highlighted and the data of Figure 3—figure supplement 1 showing this should be moved in the core of the manuscript.

5) For Figure 1C and the resulting pathways analysis, was an effect size threshold also used? Or just significance?

6) Remarkable that the endotoxemia model had so few genes differently expressed from sepsis at 24 hours (just 43). Worth a comment on the model?

7) Across which data was the functional inference done? Sepsis only? Sepsis and healthy?

8) Figure 5 should be revised and demonstrate the correlation of biomarkers with non-coding RNAs.

9) Figure 5B examines shock vs. non-shock; can Figure 1 add this subclassification?

10) We welcome the authors to present Volcano plots and analysis of non-coding RNAs between survivors and non-survivors. Since data on 1-year survival are available, can the authors shifts in expression of non-coding RNAs between original after 28-days who become survivors and non-survivors after one year?

11) Materials and methods: do any patients profiled here overlap with this group's prior microarray studies?

12) Materials and methods: why were two different microarray types used, and how was this technical difference accounted for in the sepsis-vs.-endotoxemia analysis?

13) As the results indicating correlation of non-coding RNA profiles in sepsis patients and expression profiles in the human endotoxemia model simultaneously with endotoxin tolerance is of extraordinary importance and interest, we suggest discussion of the (accidental) interplay of resistance and tolerance responses during endoxemia and sepsis.

---

## [Author Response]

Essential revisions:1) The authors might consider PMID: 29657277 (data on GEO). A very small study is here: https://doi.org/10.1089/gtmb.2017.0061. They also might consider mining their own dataset GSE65682 as others have done recently: https://translational-medicine.biomedcentral.com/articles/10.1186/s12967-020-02372-2. Many microarrays measure lncRNAs and the data are available, just not examined, online. Shouldn't the authors try to validate their findings externally?

Absolutely, we have certainly searched the public domain for independent datasets to provide a level of robustness to the patterns we observed. The three studies noted by the reviewers were indeed shortlisted in our search for potential use as external validation cohorts. However, we found certain issues with the studies that precluded their use in validation of our findings. The study by da Silva Pellegrini, et al. (PMID: 29657277) was done on a granulocyte fraction after Ficoll-Paque density gradient centrifugation of whole blood, that is, only polymorphonuclear cells were assessed. This contrasts tremendously with our study on whole blood leukocytes. The data used in the study by Dai, et al. (PMID: 28872921) is not available in the public domain, and severely underpowered since the authors performed microarray analysis of 3 septic patients relative to 3 healthy controls. The study by Cheng, et al. (PMID: 32471511), which also included our own previous data (not on non-coding RNA expression), was done by inferring the potential interaction of long non-coding RNA with coding gene expression profiles, that is, not one long non-coding RNA mentioned was measured. Thus, while of high interest, these earlier studies unfortunately cannot be used to validate our findings.

2) As a brief aside, much of the above is also true for endotoxin studies. There are also a huge number of studies looking at miRNAs in sepsis in a mechanistic fashion; nice review here: PMC6255943. All of this could be contextualized better.

We concur with the reviewers that a body of work has identified specific microRNA as potential modulators of the immune response, particularly endotoxin tolerance, as kindly noted by the reviewers in the article by Vergadi and colleagues (PMC6255943). However, interpretation of our findings in the context of previous studies on microRNA is complicated by the fact that the majority of clinical studies focused on extracellular (circulating) microRNA, whereas more mechanistic studies employed predominantly mouse macrophages or a pro-monocytic cell line THP1 or human embryonic kidney cell lines. In order to accommodate the reviewer’s suggestion, we opted to specifically discuss those miRNA that were unmasked as potential co-expression network modulators. We found that hsa-miR-200c (Figure 4E) was shown to modify the capacity of HEK293 cells for TLR-signaling in a MYD88-dependent fashion (see Wendlandt et al., 2012). We greatly appreciated the reviewers for raising this question as it has certainly helped in raising the confidence level of our network findings. The main text has been revised accordingly:

“Of note, hsa-miR-200c-3p has been shown to modify TLR4 signaling efficiency dependent on MYD88-mediated pathways in an embryonic kidney cell line (HEK293).(Wendlandt et al., 2012)”

3) Why is the analysis of non-coding RNAs in this submission accompanied by an analysis of protein-coding regions? We understand that the authors try to investigate how the expression of coding regions is modulated by non-coding regions in terms of pathways analysis, but the way data are analysed does not support such conclusions.

We analyzed protein-coding and non-coding RNA in combination because we sought to mainly infer on the functional connotations of long non-coding RNA. Functional inference of long non-coding RNA is currently limited to a handful of transcripts, unlike those of protein-coding RNA. To address this problem, we reasoned that by identifying modules (or clusters) of tightly correlating protein-coding and non-coding RNA we can then infer on functional outputs of non-coding RNA by virtue of the pathways that associate with protein-coding RNA in each module. Using this “guilt-by-association” approach we were able to position a substantial proportion of non-coding RNA with otherwise no known functional annotation. We discuss this aspect of our analysis in the Discussion section, while avoiding the topic of protein-coding RNA modulation by non-coding RNA since, as the reviewers rightly pointed out, the analytical strategy we undertook would not support it. To clarify this issue in the revised manuscript we modified the following section:

“To address this challenge, we undertook a guilt-by-association strategy that sought to position long non-coding RNA in co-expression modules of tightly correlating protein-coding RNA, thereby infer on functional outputs of long non-coding RNA by virtue of the pathways that associate with protein-coding RNA in each module.”

4) Although there is a positive association of coding-RNAs between sepsis patients and the LPS endotoxemia model, this is at the best marginal for the non-coding RNAs. This should be highlighted and the data of Figure 3—figure supplement 1 showing this should be moved in the core of the manuscript.

It is definitely remarkable that non-coding RNA expression in sepsis patients was marginally correlated to that in the human endotoxemia model. In particular, small non-coding RNA mainly showed indirect correlations. We agree with the reviewers to restructure Figure 3 by including data originally found in Figure 3—figure supplement 1. Specifically, we moved Figure 3—figure supplement 1E to Figure 3A, showing the extent of the correlation in protein-coding RNA expression between sepsis and endotoxemia.

5) For Figure 1C and the resulting pathways analysis, was an effect size threshold also used? Or just significance?

We used a fold change cutoff (≤ -1.2 or ≥ 1.2) and adjusted P-value ≤ 0.01. We included this information as a legend beneath volcano plots in Figure 1C. In addition, all other volcano plots throughout our manuscript were clarified by including legends for fold change cutoffs.

6) Remarkable that the endotoxemia model had so few genes differently expressed from sepsis at 24 hours (just 43). Worth a comment on the model?

Yes, definitely worth a comment and to accommodate the reviewer’s suggestion we have now expanded on the human endotoxemia model in the Discussion:

“The human endotoxemia model is a highly relevant in vivo model of acute systemic inflammation in the context of a controlled clinical setting (Lowry, 2005). In general, the model is characterized by a robust systemic response, including leukocyte transcriptional responses, exhibiting shared and unique temporal changes that resolve within 24 hours of bolus administration (Calvano et al., 2005; Perlee et al., 2018).”

7) Across which data was the functional inference done? Sepsis only? Sepsis and healthy?

Functional inference was done with sepsis patients only. We reasoned that in so doing we could test the co-expression modules against clinical characteristics, that is, without the confounding signal of the dramatic changes in sepsis relative to health. In order to clarify this point, we added the following to the main text:

“On the basis of a bi-weight midcorrelation matrix of the most variable protein-coding and long non-coding RNA (n=8539; coefficient of variation > 5%) in sepsis patients only, …”

8) Figure 5 should be revised and demonstrate the correlation of biomarkers with non-coding RNAs.

To address this comment, we calculated the connectivity of each transcript (sum of transcript adjacencies with respect to other module transcripts) in the co-expression network as described in the Materials and methods and illustrated in Figure 4. By ranking the intra-module connectivities we identified non-coding RNA as module “hubs”. Correlation analysis of all “hub” non-coding RNA against soluble mediators showed strong correlations, particularly for markers of systemic inflammation. Of note, these patterns fully reflected the module eigengene correlations to soluble mediators shown in Figure 5A. To illustrate these findings, we focused our attention on modules that correlate substantially to levels of different soluble mediators, that is, neutrophil degranulation (red) and TLR signaling (black). Figure 5D and E now depict dot plots of *MYOSLID* (Myocardin-Induced Smooth Muscle LncRNA, Inducer Of Differentiation) expression (red module hub) and *LUCAT1* (Lung Cancer Associated Transcript 1) expression (black module hub), respectively, against levels of inflammatory response markers IL-6, IL-8, IL-10, and acute phase response protein CRP. *MYOSLID* expression correlated directly with inflammatory markers, whereas in contrast *LUCAT1* expression was indirectly correlated. We thank the reviewers for motivating this analysis as it has provided specificity to formulating testable hypotheses on the functions of non-coding RNA thereby strengthening our study. The Results section and figure legend have been revised as follows:

“LincRNA and antisense RNA included an inducer of differentiation *MYOSLID* (Myocardin-Induced Smooth Muscle LncRNA, Inducer Of Differentiation), cell proliferation and metastasis associated antisense RNA of the titin gene (TTN-AS1) and a IL10 receptor beta subunit antisense RNA, IL10RB-AS1. […] In contrast, *LUCAT1* expression was indirectly correlated to soluble mediators of inflammation, except for CRP (Figure 5E).”

Figure 5D and E legend: “(D and E) Dot plots of (D) *MYOSLID* expression and (E) *LUCAT1* expression against soluble mediators of inflammation IL-6, IL-8 and IL-10, as well as the acute phase response protein CRP. Rho, Spearman’s coefficient.”

9) Figure 5B examines shock vs. non-shock; can Figure 1 add this subclassification?

Definitely, and since the question is related to a source of transcriptomic variation in sepsis patients, we would prefer including the analysis motivated by the reviewer in Figure 2. Figure 2 illustrates an analysis of variance in blood transcriptomes of sepsis patients against various demographic and clinical characteristics, including shock. Figure 2C and D now depict the global blood transcriptomic changes in septic shock patients relative to no shock. In the Results section we have included the following sentences to report the new findings:

“Septic shock explained low proportions of variation in RNA expression (Figure 2A), and directly comparing patients with septic shock to patients without shock resulted in 837 and 80 significantly altered protein-coding and long non-coding RNA, respectively (Figure 2C). […] Low expression protein-coding RNA included a Na+/Ca^2+^ exchanger (SLC8A1), membrane metalloendopeptidase (*MME*) and interleukin (IL-) 6 receptor (*IL6R*). Long non-coding RNA included lincRNA lung cancer associated transcript 1 (*LUCAT1*; low expression) and antisense RNA (*LRRC75A-AS1*; high expression) (Figure 2C). No significant alterations were identified in small non-coding RNA expression profiles.”

The legend of Figure 2 has also been revised to include the following details: “(C) Volcano plots depicting the changes in protein-coding and long non-coding RNA in patients discordant for septic shock on ICU admission. Horizontal (black) line denotes the adjusted p-value threshold for significance (adjusted p ≤ 0.01).”

10) We welcome the authors to present Volcano plots and analysis of non-coding RNAs between survivors and non-survivors. Since data on 1-year survival are available, can the authors shifts in expression of non-coding RNAs between original after 28-days who become survivors and non-survivors after one year?

Based on the reviewer’s comments, we firstly report results pertaining to long non-coding RNA expression in sepsis patients discordant for survival after 28 days of ICU admission. Secondly, considering only 28-day survivors we compared non-survivors to survivors after one year follow-up. We uncovered differences in protein-coding RNA profiles, albeit minimal, but no differences were observed for non-coding RNA profiles. This analysis suggests that non-coding RNA expression may not be a suitable source of signal for mortality prediction. No differences were found for 28-day survivors followed up to one year. Since our data showed no added value of RNA expression in discriminating sepsis survivors from non-survivors, we opted to only include a few sentences to report the results of the 28-day mortality analysis. However, if the reviewer would prefer us to incorporate the one year follow up analysis of 28-day survivors, we will gladly add them to the main text. The Results section was revised accordingly:

“Evaluating RNA expression in patients discordant for survival after 28 days, identified 146 significantly altered protein-coding RNA (Figure 2—figure supplement 2C). No significant differences were uncovered in non-coding RNA expression profiles, suggesting that non-coding RNA profiles obtained on ICU admission may not be suitable as mortality predictors.”

Figure 2—figure supplement 1 legend: “(C) Volcano plot of significantly altered protein-coding RNA in non-survivors relative to survivors after 28 days since ICU admission. Horizontal (black) line denotes -log10 transformed adjusted p-value thresholds.”

11) Materials and methods: do any patients profiled here overlap with this group's prior microarray studies?

None of the patients profiled here overlap with our previous microarray studies.

12) Materials and methods: why were two different microarray types used, and how was this technical difference accounted for in the sepsis-vs.-endotoxemia analysis?

Two types of arrays were used in all samples (sepsis patients and human endotoxemia) based on their specificity for measuring the different types of non-coding RNA expression. The human transcriptome array (HTA) 2.0 was designed for exon-level protein-coding RNA and mainly long non-coding RNA expression studies. The miRNA 4.1 array was used to comprehensively assay small non-coding RNA including types of small RNA (precursor miRNA, stem loop RNA, etc.) that are not covered in the HTA 2.0 microarrays. To clarify this we added the following to the Materials and methods section:

“Both arrays were done on all samples (sepsis patients, controls and healthy subjects injected with LPS)”.

13) As the results indicating correlation of non-coding RNA profiles in sepsis patients and expression profiles in the human endotoxemia model simultaneously with endotoxin tolerance is of extraordinary importance and interest, we suggest discussion of the (accidental) interplay of resistance and tolerance responses during endoxemia and sepsis.

In accordance with the suggestion of the reviewer we added the following to the Discussion:

“The host response during infection is characterized by a balance between resistance (seeking to limit the pathogen load) and tolerance (aiming to retain cell and organ functions)(Schneider and Ayres, 2008). […] While our time-sequential data in healthy humans injected with LPS suggest that coding and long non-coding RNA profiles in blood leukocytes of sepsis patients particularly reflect a tolerant state, time course studies in patients are needed to increase the insight into the role of distinct RNA species in the interplay between resistance and tolerance”.